# Prevalence of timely complementary feeding initiation and associated factors among mothers having children aged 6–24 months in rural north-central Ethiopia: Community based cross-sectional study

**Moges Wubneh Abate**[1]*, **Adane Birhanu Nigat**[1], **Agimasie Tigabu Demelash**[1], **Tigabu Desie Emiru**[2], **Nigusie Selomon Tibebu**[2], **Chalie Marew Tiruneh**[2], **Amsalu Belete**[3], **Tilahun Kegne Abebe**[4], **Moges Yinges Yitayew**[5]

1 Department of Adult Health Nursing, College of Health Sciences, Debre Tabor University, Debre Tabor, Ethiopia, 2 Department of Pediatrics and Child Health Nursing, College of health sciences, Debre Tabor-University, Debre Tabor, Ethiopia, 3 Department of Mental Health, School of medicine, Debre Tabor University, Debre Tabor, Ethiopia, 4 Department of Public Health, Debre Tabor Comprehensive Specialised Hospital, Debre Tabor, Ethiopia, 5 Department of Public Health, Ethiopian Red Cross-Society, Debre Tabor, Ethiopia

* wmoges7@gmail.com

## Abstract

### Background

Complementary foods are defined as any solid or liquid foods other than breast milk offered to children. Timely initiation of complementary foods during infancy is necessary for growth and development. The first two years of life are important period for rapid physical, cognitive and social development that requires optimal nutrition. Currently, there is no study done in this rural community about timely initiation of complementary feeding.

### Objective

The main aim of this study was to assess the prevalence of timely initiation of complementary feeding among mothers having children aged 6–24 months in Farta district, rural Ethiopia.

### Methods

A community-based cross-sectional study was employed from December 2020 to February 2021 among 570 mothers by using multi-stage sampling techniques. Data were collected using a structured interviewer-administered questionnaire and entered into Epi Data 4.6 then transferred to Statistical Package for Social Science version 25 for analysis. Bivariate and multivariable logistic regression analysis with a 95% confidence interval carried out to determine the association between explanatory and the outcome variables. A P-value of < 0.05 was considered statistically significant.

**Data Availability Statement:** All data are available at https://doi.org/10.6084/m9.figshare.c.5940700.v2.

**Funding:** The author(s) received no specific funding for this work.

**Competing interests:** The authors have declared that no competing interests exist.

## Results

The prevalence of timely initiation of complementary feeding among mothers having children aged 6–24 months was 51.9%. Institutionaldelivery [(AOR = 2.10, 95% CI: (1.31–3.32)],Keeping livestock [(AOR = 2.21, 95% CI: (1.35, 3.65)], Postnatal follow up [(AOR = 0.60, 95% CI:(0.36, 0.77)],merchants [(AOR = 4.58; 95% CI:1.99, 10.55)], and daily labourer [(AOR = 2.88, 95% CI:(1.50–5.51)] were statistically associated with timely initiation of complementary feeding.

## Conclusion

This finding revealed that the prevalence of timely initiation of complementary feeding is still low. Factors affecting timely initiations of complementary feeding were home delivery, unable to attend postnatal care follow-up, being housewife and farmers. All health professionals including health extension workers should give special attention to advising and counseling for mothers and their husbands about timely initiation of complementary feeding.

## Introduction

### Background

Complementary foods are defined as any solid or liquid foods other than breast milk, offered to infants [1]. The initiations of solid foods for an infant is known as complementary feeding [2]. Initiations of additional or complementary foods at six months are essential for growth and development of the children [3]. Exclusive breastfeeding for six months should charge by complementary feeding such as offering solid foods [4]. The world health organization (WHO) is recommend exclusive breastfeeding for the first six months of life and mothers should initiate soft, semi-solid, or solid food for their infants at the age of six months [5]. The first two years of life are important and "critical window" for rapid physical, cognitive and social development that requires optimal nutrition [6, 7].

In 2017, the world prevalence of timely initiation of complementary feeding was 64.5% [8].

In the world, near to eleven million infants and young children death reported annually due to the absence of timely initiations of complementary feeding [9].

In developing countries, over two-hundred million children under five years were unable to reach their capacity in cognitive development because of not initated timely complementary feeding [10]. The immediate consequences of poor nutrition during early childhood are increased the chance of morbidity, mortality and delayed mental development [11].

Ethiopia is still faced, undernourished children. This is mainly due to the absence of timely complementary feeding. Currently, there is no study done in rural community of Ethiopia including the study area about timely initiation of complementary feeding. Therefore, this study will be used as a source of information for the policymakers to improve timely initiation to complementary feeding. It also used as baseline data for further research. The main purpose of this study was to assess timely initiation of complementary feeding among mothers having children aged 6–24 months in Farta district, Rural Ethiopia.

## Materials and methods

### Study area and period

The study was conducted in Farta district, which is one of the 13 districts found in South Gondar zone, Amhara, Ethiopia. It has 36 rural kebeles. There are 8 health centres and 36 health posts providing health service for the district population. The total population of Farta district is 250,731. Of these total populations, 17,664 children were under two years of age.

### Study design and source population

A community-based cross-sectional study was conducted among mothers who have Children 6–24 months of age in Farta district, South Gondar zone, Northcentral Ethiopia.

### Inclusion criteria

All mothers who were had children aged from six months to two years in Farta district.

### Exclusion criteria

Mothers who were seriously ill and unable to provided information during data collection period.

### Study variables

**Dependent variable.**   Timely initiation of complementary feeding.
**Independent variables.**   **Socio-demographic and economic characteristics** (Age of mothers, occupation, educational level, read magazines and income).
**Maternal reproductive history and health service utilization** (ANC follow up, number of visits, PNC follow up, place of delivery, number of live birth, birth interval, number of children delivered at a time).

### Operational definition

**Timely initiation of complementary feeding.**   It is the initiation of additional foods for a young children at sixth months of age along with continuing breastfeeding [12].

### Sample size, sampling techniques and procedure

By taking the prevalence of timely initiation of complementary feeding 34% [13]. Epi info version 7.2.2.6 was used to calculate the sample by considered confidence interval 95%, power 80% with design effect 1.5. Based on this the sample size was 518. By adding a 10% non-response rate the final sample size was 570. A multi-stage sampling technique was used to select the study participants. First, 8 out of 36 Farta districts were selected randomly using lottery method. There were 2198 households in the selected kebeles having under-two-years-old children. The calculated sample size was proportionally allocated to the selected kebeles based on the number of households. In the second stage, a systematic random sampling technique was used to selected mothers at each selected kebeles. The sampling frame was obtained from the health extension workers at each selected kebeles. Finally, proportionally allocated study participants were selected by using systematic random sampling every 4th interval while the first household was selected by lottery method.

### Data collection tool

Structured interviewer-administered questionnaires were used to collect the timely initiation of complementary feeding and its associated factors. The questionnaire was adapted from key indicators recommended by WHO and IYCF strategy of Ethiopia [12, 14]. The questionnaire

composed of socio-demographic and economic characteristics, health care, and complementary feeding practice-related variables. It was initially prepared in English and then translated from English to Amharic and re-translated back to English by a language expert to maintain its consistency. Pre-testing was done on 5% of mothers at Ebinat district to check the reliability of the tool. Data were collected by eight health extension workers and two masters nurse supervisors for monitoring the overall data collection process. One day of training was given to data collectors and supervisors.

## Data quality assurance, processing and analysis

Adequate training was provided for eight data collectors and two supervisors. The codes were given to the questionnaire. Data collectors and supervisors checked the filled questionnaire for completeness every day. Problems encountered during the study period were discussed in the study team and solved. Subsequently, the data were entered using Epi Data 4.6. The generated data was exported to a statistical package for social sciences (SPSS) version 25. The data was cleaned by visualizing, calculating frequencies, and sorting. The analysis was done with descriptive statistics by using frequency and percentage while bivariate and multivariable regression analysis between dependent and independent variables were performed using binary logistic regression. Hosmer and Lemeshow test was checked for model goodness of fit. During the bivariate analysis0.2 p-values were taken into multivariable regression. A 95% confidence interval (CI) and a P-value of less than 0.05 was taken as a significant association. Results were presented in text and tables.

## Ethics approval and consent to participate

Ethical approval was obtained from ethical review board of (blind for peer review) with a protocol number **00310/2020**. A formal permission letter was communicated at each level. The participants were well informed about the purpose, the right to refuse and participate in the study. Finally, data was collected after oral and written consent from each respondent.

## Results

### Socio-demographic and economic characteristics

From a total of 570 samples, 536 mothers were participated in the study with a response rate of 94%. Regarding educational status respondents of the respondents,266 (49.6%) were able to read and write (Table 1).

### Maternal reproductive history and health service utilization

Among the respondents, 315 (58.8%) of the mothers were primiparous. More than half of the mothers 313(58.4%) had ANC follow up (Table 2).

### Timely initiation of complementary feeding

From the total participants 278 (51.9%) of mothers were initiated complementary feeding at six months.

### Factors associated with timely initiation of complementary feeding

In the multivariable logistic regression analysis, occupations of the mothers, occupation of the spouse, Place of current delivery and PNC follow up were significantly associated with timely initiation of complementary feeding. Those mothers who had not postnatal care follow-up

**Table 1. Socio-demographic characteristics of mothers in Farta district (n = 536), Ethiopia, 2021.**

| Variables | Category | Frequency (N) | Percent (%) |
|---|---|---|---|
| Age of mother in years | 15–19 | 36 | 6.7 |
| | 20–25 | 157 | 29.3 |
| | 26–29 | 90 | 16.8 |
| | > = 30 | 253 | 47.2 |
| Educational status of mother | Unable to read and write | 115 | 21.5 |
| | able to read and write | 266 | 49.6 |
| | Primary | 86 | 16.0 |
| | Secondary | 22 | 4.1 |
| | Diploma and above | 47 | 8.8 |
| Educational status of spouse | Unable to read and write | 131 | 24.4 |
| | able to read and write | 217 | 40.5 |
| | Primary | 144 | 26.9 |
| | Secondary education | 21 | 3.9 |
| | Diploma and above | 23 | 4.3 |
| Occupation of mother | Housewife | 275 | 51.3 |
| | Keep livestock | 121 | 22.6 |
| | Daily labourer | 83 | 15.5 |
| | Merchant | 26 | 4.9 |
| | Governmental employee | 31 | 5.8 |
| Occupation of spouse | Farmer | 405 | 75.6 |
| | Governmental employee | 20 | 3.7 |
| | Merchant | 53 | 9.9 |
| | daily labourer | 58 | 10.8 |
| Read magazines and books (at least one month) | Yes | 41 | 7.6 |
| | No | 495 | 92.4 |
| Monthly income of the household | ≤500 ETB | 192 | 35.8 |
| | 501–1000 ETB | 170 | 31.7 |
| | 1001–1500 ETB | 4 | 0.7 |
| | 1501–2000 ETB | 87 | 16.2 |
| | ≥2001 ETB | 83 | 15.5 |
| Family size in number | 2–3 family | 404 | 75.4 |
| | 4–6 family | 103 | 19.2 |
| | > 6 family | 29 | 5.4 |

**Abbreviation**: ETB = Ethiopian Birr.

were 40% less likely to initiate timely complementary feeding. Mothers had merchant husbands were 4.58 times more likely to initiate timely complementary feeding than mothers whose husband were farmers [(AOR = 4.58; 95% CI: (1.99, 10.55)] (Table 3).

## Discussions

The main reason to conduct this study was to assess timely initiation of complementary feeding in Farta district, rural Ethiopia. Mothers who had children 6–24 months were the study participant.

The result of this study revealed that the prevalence of timely initiation of complementary feeding was 51.9% [(95% CI; (48%, 56%))]. The timely initiation of complementary feeding

**Table 2. Maternal reproductive history and health service utilization of mothers in Farta district, Ethiopia, 2020.**

| Variables | Categories | Frequency (N = 536) | Percent (%) |
|---|---|---|---|
| Number of live births | ≤ 2 | 315 | 58.8 |
|  | 3–4 | 185 | 34.5 |
|  | ≥ 5 | 36 | 6.7 |
| Birth interval | ≤ 2 years | 273 | 50.9 |
|  | ≥ 3 years | 263 | 49.1 |
| ANC follow up | Yes | 313 | 58.4 |
|  | No | 223 | 41.6 |
| Number of ANC follow up | 1–2 times | 159 | 29.7 |
|  | ≥ 3 times | 154 | 28.7 |
| During ANC visit told time to CF | Yes | 265 | 68 |
|  | No | 123 | 32 |
| Place of current delivery | Home | 119 | 22.2 |
|  | Institutional | 417 | 77.8 |
| PNC follow up | Yes | 262 | 48.9 |
|  | No | 274 | 51.1 |
| During PNC visit told time to CF | Yes | 200 | 37.3 |
|  | No | 62 | 11.6 |
| Number of children delivered. | 1 | 432 | 80.6 |
|  | ≥ 1 | 104 | 19.4 |

**Abbreviations**: ANC = Antenatal care, CF = Complementary Feeding, PNC = Postnatal Care.

**Table 3. Factors associated with timely initiation of complementary feeding among mothers who had 6–24 months children in farta district.**

| Variables | Categories | Timely initiation complementary feeding | | COR (95% CI) | AOR (95%CI) |
|---|---|---|---|---|---|
|  |  | No | Yes |  |  |
| participant occupation | Housewife | 154 | 121 | 1 | 1 |
|  | Keep livestock | 34 | 87 | 3.26 (2.05–5.17)** | 2.21(1.33–3.68)** |
|  | Daily labourer | 45 | 38 | 1.08 (0.66–1.76) | 0.79 (0.46–1.36) |
|  | Merchant | 8 | 18 | 2.86 (1.20–6.81) * | 1.02 (0.32–3.28) |
|  | Employee | 17 | 14 | 1.05 (0.50–2.21) | 0.97 (0.42–2.22) |
| Occupation of the spouse | Farmer | 221 | 184 | 1 | 1 |
|  | Employee | 13 | 7 | 0.65 (0.253–1.56) | 0.54 (0.20–1.45) |
|  | Merchant | 8 | 45 | 6.76(3.11–14.70)** | 4.58(1.99–10.55)** |
|  | Daily labourer | 16 | 42 | 3.15 (1.72–5.79)** | 2.88 (1.50–5.51) ** |
| Place of delivery | Home | 75 | 44 | 1 | 1 |
|  | Institutional | 183 | 234 | 2.18 (1.43–3.31)** | 2.10(1.31–3.32)** |
| PNC follow up | Yes | 104 | 158 | 1 | 1 |
|  | No | 154 | 120 | 0.51 (0.36–0.72)** | 0.60(0.41–0.89)** |

**Note**: 1 = Reference group,

*P≤ **0.05** = statistically significant,

**P≤ **0.01** = highlystatistically significant

Hosmer-Lemeshow goodness of fit test = 0.57

**Abbreviations**: AOR = Adjusted odd ratio, CI = Confidence interval, COR = Crude odd ratio

practice in this study was higher than findings from Iran(44.8%) [15], Kamba district, Ethiopia (40.4%) [16]. This difference is because of in Iran the study participants were those mothers had children up to two and a half year of life which may prone to recall bias compared to our study. Besides this in Iran low birth weight and multiple birth children were excluded in the study this could also underestimate the finding. The possible reason for variation with Kamba Woreda may be due to difference in the study period since the study in Kamba was done before six year ago and nowadays, the health extension workers play a great role in the dissemination of information about child feeding practice for the community. The finding was lower than the study conducted in Bangladesh (71%) [17], Halaba Kulito (57.8%) [18], Dessie referral hospital(65.1%) [19] and Addis Ababa (83%) [20], Hiwot Fana specialized hospital (60.5%) [21], Mekelle (62.8%) [22], Lalibela (63%) [23], Sodo town, Southern Ethiopia (71.2%) [24]. These incongruent findings are may be due to timely initiation of complementary feeding in Bangladesh included those mothers who initiated feeding in 6–8 months compared with our study exactly at 6 months. Also differences in Addis Ababa, Hiwot Fana hospital, Mekelle and Lalibela were done both in an urban and rural community while our study solely conducted in a rural community may bring this great variation because mothers in urban community are near for accessing information by different media regarding timely initiation of complementary feeding compared to a rural community. The discripances with the study done in Sodo town, Ethiopia may be due to variations in residence, since a study done in Sodo was in urban community this may give better opportunity for timely initiation of complementary feeding while, our study done in rural community may have negative impact by different mechanisms like the community may not be accessed adequate information about timely initiation of complementary feeding.

Occupation of mothers, occupation of the spouses, place of delivery and PNC visits were associated with timely initiation of complementary feeding.

Occupation of the mother is one of the factors that affect timely initiations of complementary feeding. In this case, those mothers who kept livestock were 2.21 times more likely to initiated timely complementary feeding than a housewife. This is consistent with studies conducted in Kamba and Lalibela districts [16, 23]. The reason may be due to keep livestock mothers usually stay out of home and they may get sufficient information about timely initiation of complementary feeding in contrast to housewife as they stay at home to wean their children early or late initiation of complementary feeding.

The occupations of the spouse were also affect timely initiation of complementary feeding. Daily labourer and merchants were more likely to initiated timely complementary feeding compared to farmers. This finding was consistent with a study done in Pakistan [25], Gode district, Ethiopia [26]. This may be happened since merchants has better income and able to provide complementary feeding to their young infants. Beside this merchants and daily labourers may also have better chance of getting information regarding complementary feeding compared with farmers.

Postnatal care user mothers' were more likely to initiated timely complementary feeding to their infants compared to those who did not follow the service. The finding is inline with Axum, Ethiopia [27] and Kamba district, Ethiopia [16]. Mothers who had no postnatal care follow up were start complementary feeding earlier before 6 months or later after 8 months compared to mothers who not followed the PNC care. A postnatal period could be an ideal time to counsel mothers on optimal complementary feeding practiceas studies shown in Mekelle, Ethiopia [22], Addis Ababa [20] and Lalibela District, Ethiopia [23].

Generally, this study has a great impact or benefit for mothers to enhancing their habit of timely initiation of complementary feeding and also helps to reducing malnutrions of the children. Since, there is no study done in rural community of Ethiopia including the study area,

this study helps to show the current conditions or figures of complementary feeding for different stakeholders. Finally, this study was mainly focused on the prevalence of timely initiation of complementary feeding. So that types of foods items or food diversities to be initiated were not addressed in this study.

## Conclusion

The prevalence of timely introduction of complementary feeding in children aged 6–24 months is low in study setting. This finding was mainly affected by home delivery, unable to attend postnatal care follow-up, being housewife and farmers. This indicates that still, the community did not get adequate information about timely initiation of complementary feeding. It needs great effort and collaborative work from different stakeholders; like ministrey of health, regional health bureau, district health office, non governmental organizations and health care professionals to disseminateing information about the importance of timely initiation of complementary feeding to overcome child malnutrition. All health professionals including health extension workers should give attention to advising and counseling for mothers and their husbands about timely initiation of complementary feeding during antenatal, delivery and post natal period. Further research should be conducted by using qualitative study design to understand deeply socio-cultural and behavioral related factors towards timely initiation of complementary feeding.

## Supporting information

**S1 File.**
(DOCX)

## Acknowledgments

We would like to acknowledge all data collectors for the success of our work.

## Author Contributions

**Conceptualization:** Moges Wubneh Abate, Adane Birhanu Nigat, Agimasie Tigabu Demelash, Tigabu Desie Emiru, Nigusie Selomon Tibebu, Chalie Marew Tiruneh, Amsalu Belete, Tilahun Kegne Abebe, Moges Yinges Yitayew.

**Data curation:** Moges Wubneh Abate, Adane Birhanu Nigat, Agimasie Tigabu Demelash, Tigabu Desie Emiru, Nigusie Selomon Tibebu, Chalie Marew Tiruneh, Amsalu Belete, Tilahun Kegne Abebe, Moges Yinges Yitayew.

**Formal analysis:** Moges Wubneh Abate, Adane Birhanu Nigat, Agimasie Tigabu Demelash, Tigabu Desie Emiru, Nigusie Selomon Tibebu, Chalie Marew Tiruneh, Amsalu Belete, Tilahun Kegne Abebe, Moges Yinges Yitayew.

**Funding acquisition:** Moges Wubneh Abate, Adane Birhanu Nigat, Agimasie Tigabu Demelash, Tigabu Desie Emiru, Nigusie Selomon Tibebu, Chalie Marew Tiruneh, Amsalu Belete, Tilahun Kegne Abebe, Moges Yinges Yitayew.

**Investigation:** Moges Wubneh Abate, Adane Birhanu Nigat, Agimasie Tigabu Demelash, Tigabu Desie Emiru, Nigusie Selomon Tibebu, Chalie Marew Tiruneh, Amsalu Belete, Tilahun Kegne Abebe, Moges Yinges Yitayew.

**Methodology:** Moges Wubneh Abate, Adane Birhanu Nigat, Agimasie Tigabu Demelash, Tigabu Desie Emiru, Nigusie Selomon Tibebu, Chalie Marew Tiruneh, Amsalu Belete, Tilahun Kegne Abebe, Moges Yinges Yitayew.

**Project administration:** Moges Wubneh Abate, Adane Birhanu Nigat, Agimasie Tigabu Demelash, Tigabu Desie Emiru, Nigusie Selomon Tibebu, Chalie Marew Tiruneh, Amsalu Belete, Tilahun Kegne Abebe, Moges Yinges Yitayew.

**Resources:** Moges Wubneh Abate, Adane Birhanu Nigat, Agimasie Tigabu Demelash, Tigabu Desie Emiru, Nigusie Selomon Tibebu, Chalie Marew Tiruneh, Amsalu Belete, Tilahun Kegne Abebe, Moges Yinges Yitayew.

**Software:** Moges Wubneh Abate, Adane Birhanu Nigat, Agimasie Tigabu Demelash, Tigabu Desie Emiru, Nigusie Selomon Tibebu, Chalie Marew Tiruneh, Amsalu Belete, Tilahun Kegne Abebe, Moges Yinges Yitayew.

**Supervision:** Moges Wubneh Abate, Adane Birhanu Nigat, Agimasie Tigabu Demelash, Tigabu Desie Emiru, Nigusie Selomon Tibebu, Chalie Marew Tiruneh, Amsalu Belete, Tilahun Kegne Abebe, Moges Yinges Yitayew.

**Validation:** Moges Wubneh Abate, Adane Birhanu Nigat, Agimasie Tigabu Demelash, Tigabu Desie Emiru, Nigusie Selomon Tibebu, Chalie Marew Tiruneh, Amsalu Belete, Tilahun Kegne Abebe, Moges Yinges Yitayew.

**Visualization:** Moges Wubneh Abate, Adane Birhanu Nigat, Agimasie Tigabu Demelash, Tigabu Desie Emiru, Nigusie Selomon Tibebu, Chalie Marew Tiruneh, Amsalu Belete, Tilahun Kegne Abebe, Moges Yinges Yitayew.

**Writing – original draft:** Moges Wubneh Abate, Adane Birhanu Nigat, Agimasie Tigabu Demelash, Tigabu Desie Emiru, Nigusie Selomon Tibebu, Chalie Marew Tiruneh, Amsalu Belete, Tilahun Kegne Abebe, Moges Yinges Yitayew.

**Writing – review & editing:** Moges Wubneh Abate, Adane Birhanu Nigat, Agimasie Tigabu Demelash, Tigabu Desie Emiru, Nigusie Selomon Tibebu, Chalie Marew Tiruneh, Amsalu Belete, Tilahun Kegne Abebe, Moges Yinges Yitayew.

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
