## [Decision Letter · Decision Letter 0]

15 Dec 2021

PONE-D-21-34084Prevalence of Timely Complementary Feeding Initiation and Associated Factors among Mothers Having Children Aged 6-24 Months in Rural North-central Ethiopia: Community based Cros-sectional StudyPLOS ONE

Dear Dr. ABATE,

Thank you for submitting your manuscript to PLOS ONE. After careful consideration, we feel that it has merit but does not fully meet PLOS ONE’s publication criteria as it currently stands. Therefore, we invite you to submit a revised version of the manuscript that addresses the points raised during the review process.

Please note that further language improvement is highly indicated in the manuscript to be considered as a sound one. Consider revising the spelling, grammar, diction, and syntax throughout the manuscript for increased clarity to meet the standards for PLOS one publication. 

We look forward to receiving your revised manuscript.

Kind regards,

Ammal Mokhtar Metwally, Ph.D (MD)

Academic Editor

PLOS ONE

Journal Requirements:

Reviewers' comments:

Reviewer's Responses to Questions

**Comments to the Author**

1. Is the manuscript technically sound, and do the data support the conclusions?

Reviewer #1: Yes

Reviewer #2: Partly

Reviewer #3: Yes

2. Has the statistical analysis been performed appropriately and rigorously? 

Reviewer #1: Yes

Reviewer #2: No

Reviewer #3: I Don't Know

3. Have the authors made all data underlying the findings in their manuscript fully available?

Reviewer #1: Yes

Reviewer #2: Yes

Reviewer #3: No

4. Is the manuscript presented in an intelligible fashion and written in standard English?

Reviewer #1: Yes

Reviewer #2: No

Reviewer #3: No

5. Review Comments to the Author

Reviewer #1: The whole manuscript is based on a community-based cross-sectional study to discuss the prevalence of complementary feeding initiation and associated factors

among mothers having children aged 6-24 Months in rural north-central Ethiopia. The investigation focus on the mother's situation, such as mother’s career, spouse career, mother's age, work environment, birth intervals.

Major critique:

1) Authors claimed that it needs great effort to disseminate information regarding the importance of timely initiation of complementary feeding to overcome child malnutrition. But there have been prospective studies on related topics, which may put this article in a insufficient innovation position, such as Complementary Feeding Habits in Children Under the Age of 2 YearsLiving in the City of Adama in theOromia Region in Central Ethiopia:Traditional Ethiopian Food Study, 2021.

Other points

2) To explore the prevalence, it would be helpful to discuss th status of children's growth and some specific diseases in the study, such as wasting, stunting, underweight, and low body mass index.

3) The specific food of complementary feeding may add into the study, Including but not limited to vegetables, cereals (teff, wheat, barley), and fruits.

4) The discussion focus comparison between this study and the others’ work, but lack for explaining the innovation and advantage, or their own defects.

Reviewer #2: Dear Authors

Thanks for the efforts that has been made to accomplish this work. The topic is interesting, however, the selected test is not appropriate to report such results. I would highly suggest asking for consultation to re do the test especially for the factor analysis section. Grammatical check is needed as well and you need to remove some parts or re write it such as dependent and independent variables, operational definition etc.

Reviewer #3: The data analysis is incomplete and can be expanded further. There is lack of novelty in the study and researchers should try to explore more variables which may have a relation with the timely initiation of complementary feeding. The overall language needs editing. The flow is not maintained.

6. PLOS authors have the option to publish the peer review history of their article (what does this mean?). If published, this will include your full peer review and any attached files.

Reviewer #1: No

Reviewer #2: No

Reviewer #3: **Yes: **Nishtha Kathuria

---

## [Author Response · Author response to Decision Letter 0]

7 Jan 2022

Response to Reviewers

 PONE-D-21-34084

Prevalence of Timely Complementary Feeding Initiation and Associated Factors among Mothers Having Children Aged 6-24 Months in Rural North-central Ethiopia: Community based Cros-sectional Study

Dear Dr. Ammal Mokhtar Metwally, Ph.D (MD)

Thank you very much for your ongoing consideration of our manuscript (PONE-D-21-34084) for publication in PLOS ONE. We have revised the manuscript to reflect your (editor) comments and those of the reviewers, and our point-by-point responses are shown on the next few pages. We appreciate the time spent by you and the reviewers, and hope you agree that the revised manuscript is both improved and now suitable for publication. We look forward to hearing from you at your earliest convenience. 

Yours sincerely,

Moges Wubneh Abate

Email:wmoges7@gmail.com

Editor comments

1. Background: The timely initiation of complementary foods during infancy is necessary for growth and development. In the past years, despite efforts put to increase timely initiation of complementary feeding among lactating mothers in Ethiopia by different stakeholders, the goal is not attain as expected----not clear

Authors’ response

Thank you dear editor for your insight and we have amended this section. It now reads:

Background: Complementary foods are defined as any solid or liquid foods with a nutritional value other than breast milk offered to children. Timely initiation of complementary foods during infancy is necessary for growth and development. The first two years of life are important period for rapid physical, cognitive and social development that requires optimal nutrition. 

2. Introduction needs overall improvement and language editing

Authors’ response

Thank you dear editor for your insight and we have amended this section. It now reads in page4 (paragraph1; line2-10, paragraph 2; line1-4, paragraph3; line1-4 and paragraph4; line2-5).

3. It would be good to depict the relation between mothers reproductive history and health service utilization with initiation of complementary food. 

Authors’ response

Dear editor we accepted your comment, but the authors try to showed the assosation between some of reproductive history like partiy and health service utilizations like ANC, PNC history with initiation of complementary feeding even if; some of these variables had not significant association with timely initiation of complementary feeding.

Reviewer #1: The whole manuscript is based on a community-based cross-sectional study to discuss the prevalence of complementary feeding initiation and associated factors among mothers having children aged 6-24 Months in rural north-central Ethiopia. The investigation focus on the mother's situation, such as mother’s career, spouse career, mother's age, work environment, birth intervals.

Major critique:

1) Authors claimed that it needs great effort to disseminate information regarding the importance of timely initiation of complementary feeding to overcome child malnutrition. But there have been prospective studies on related topics, which may put this article in a insufficient innovation position, such as Complementary Feeding Habits in Children Under the Age of 2 Years Living in the City of Adama in theOromia Region in Central Ethiopia: Traditional Ethiopian Food Study, 2021.

Other points

2) To explore the prevalence, it would be helpful to discuss th status of children's growth and some specific diseases in the study, such as wasting, stunting, underweight, and low body mass index.

3) The specific food of complementary feeding may add into the study, Including but not limited to vegetables, cereals (teff, wheat, barley), and fruits.

4) The discussion focus comparison between this study and the others’ work, but lack for explaining the innovation and advantage, or their own defects.

Authors’ response for Reviewer #1

Authors’ response: 1. 

Dear reviewer, thank you for your insight but the objective of the study done in Adama city was focused on food habits or nutritional status of the children were assessed at the same time where as, our study is focused on howmany of mothers were initiated complementary feeding at 6 months

Authors’ response: 2

Dear reviewer thank you for your suggestions but the objective of this study was timely initiations of complementary feeding, not the nutritional status of the children. There are studies done specfically to address nutritional status among children. Example: Undernutrition and associated factors among urban children aged 24–59 months in Northwest Ethiopia: a community based cross sectional study

Authors’ response: 3

Thank you dear reviewer for constructive comment but this study was mainly focused does mothers in rural community initiated timely complementary feeding irrespective of food diversity. And also the authors set reccommendation for other researchers in discussion section to include food items page 15; paragraph 5, line 5-7.

Authors’ response: 4

Thank you dear reviewer for constructive comments and the authors corrected the comments. Now reads it on page 15; paragraph 5, line 1-7.

Reviewer #2: Dear Authors

Thanks for the efforts that has been made to accomplish this work. The topic is interesting; however, the selected test is not appropriate to report such results. I would highly suggest asking for consultation to re do the test especially for the factor analysis section. Grammatical check is needed as well and you need to remove some parts or re write it such as dependent and independent variables, operational definition etc.

Authors’ response for Reviewer #2

Authors’ response

Dear reviewer thank you for your kind words sothat, based on your suggestion we were ask consultation to different researchers. So based on the resarchers recommendation for this study is pereferable to be binary logistic regression test. Grammar errors were checked and corrected accordingly through out the entire manuscripts based on your recommendation. We re-write and amended in dependent and independent variables, operational definition based on reviewer recommandations. Now read it on page 5 and 6 in the manuscript.

Reviewer #3: The data analysis is incomplete and can be expanded further. There is lack of novelty in the study and researchers should try to explore more variables which may have a relation with the timely initiation of complementary feeding. The overall language needs editing. The flow is not maintained.

Authors’ response for Reviewer #3

Authors’ response

Dear reviewer thank you for your comment the objective of this study is mainly focused on howmany of mothers in this rural community of Ethiopia were initiated complementary feeding. Eventhough, there are unexplored variables present in this study; the authors believed that the tested variables are adequate. We have also set recommendation for other researchers to conducting study in this area or tittle in last paragraph of discussion section. The language or grammar errors were corrected as per recommendations of the reviewer in the entire section of the manuscript. We have made amendement on the manuscript flow in the entire section based on your recommendation.

---

## [Decision Letter · Decision Letter 1]

1 Apr 2022

Prevalence of Timely Complementary Feeding Initiation and Associated Factors among Mothers Having Children Aged 6-24 Months in Rural North-central Ethiopia: Community based Cros-sectional Study

PONE-D-21-34084R1

Dear Dr. ABATE,

We’re pleased to inform you that your manuscript has been judged scientifically suitable for publication and will be formally accepted for publication once it meets all outstanding technical requirements.

Kind regards,

Ammal Mokhtar Metwally, Ph.D (MD)

Academic Editor

PLOS ONE

Additional Editor Comments (optional):

A great effort was made by the authors to utilize the feedback that was provided for them to correct for resubmission

Reviewers' comments:

Reviewer's Responses to Questions

**Comments to the Author**

1. If the authors have adequately addressed your comments raised in a previous round of review and you feel that this manuscript is now acceptable for publication, you may indicate that here to bypass the “Comments to the Author” section, enter your conflict of interest statement in the “Confidential to Editor” section, and submit your "Accept" recommendation.

Reviewer #2: All comments have been addressed

2. Is the manuscript technically sound, and do the data support the conclusions?

Reviewer #2: Yes

3. Has the statistical analysis been performed appropriately and rigorously? 

Reviewer #2: Yes

4. Have the authors made all data underlying the findings in their manuscript fully available?

Reviewer #2: Yes

5. Is the manuscript presented in an intelligible fashion and written in standard English?

Reviewer #2: Yes

6. Review Comments to the Author

Reviewer #2: Dear Authors

Thanks for considering all comments.

This version is much better than previous one.

I wish you the good luck in your submission this time.

If you need any further clarification please do contact me

Cheers

7. PLOS authors have the option to publish the peer review history of their article (what does this mean?). If published, this will include your full peer review and any attached files.

Reviewer #2: **Yes: **Manal I. Kassab

---

## [Editor Report · Acceptance letter]

9 May 2022

PONE-D-21-34084R1 

Prevalence of Timely Complementary Feeding Initiation and Associated Factors among Mothers Having Children Aged 6-24 Months in Rural North-central Ethiopia: Community based Cros-sectional Study 

Dear Dr. Abate:

I'm pleased to inform you that your manuscript has been deemed suitable for publication in PLOS ONE. Congratulations! Your manuscript is now with our production department. 

Kind regards, 

on behalf of

Professor Ammal Mokhtar Metwally 

Academic Editor

PLOS ONE